# CRISPR-Cas9 Technology for the Creation of Biological Avatars Capable of Modeling and Treating Pathologies: From Discovery to the Latest Improvements

**DOI:** 10.3390/cells11223615

**Published:** 2022-11-15

**Authors:** Ali Nasrallah, Eric Sulpice, Farah Kobaisi, Xavier Gidrol, Walid Rachidi

**Affiliations:** Univ. Grenoble Alpes, CEA, INSERM, 38000 Grenoble, France

**Keywords:** CRISPR-Cas9, disease modeling, gene therapy, genome editing

## Abstract

This is a spectacular moment for genetics to evolve in genome editing, which encompasses the precise alteration of the cellular DNA sequences within various species. One of the most fascinating genome-editing technologies currently available is Clustered Regularly Interspaced Palindromic Repeats (CRISPR) and its associated protein 9 (CRISPR-Cas9), which have integrated deeply into the research field within a short period due to its effectiveness. It became a standard tool utilized in a broad spectrum of biological and therapeutic applications. Furthermore, reliable disease models are required to improve the quality of healthcare. CRISPR-Cas9 has the potential to diversify our knowledge in genetics by generating cellular models, which can mimic various human diseases to better understand the disease consequences and develop new treatments. Precision in genome editing offered by CRISPR-Cas9 is now paving the way for gene therapy to expand in clinical trials to treat several genetic diseases in a wide range of species. This review article will discuss genome-editing tools: CRISPR-Cas9, Zinc Finger Nucleases (ZFNs), and Transcription Activator-Like Effector Nucleases (TALENs). It will also encompass the importance of CRISPR-Cas9 technology in generating cellular disease models for novel therapeutics, its applications in gene therapy, and challenges with novel strategies to enhance its specificity.

## 1. Introduction

Eukaryotic genomes comprise billions of DNA nitrogenous bases. The ability to change certain DNA nitrogenous bases at accurately defined locations is crucial in molecular genetics, biotechnology, and medicine [1]. Genetic modifications have long been a matter of concern in molecular biology research. Nowadays, investigating the fundamental function of genes affecting human phenotypes is a big task for researchers to confront. A simple example to understand the role of a particular gene is to either overexpress or switch it off and analyze the outcomes, which are time consuming, challenging, and expensive [2,3]. The introduction of the genome-editing concept to the research field paved the way for functional genomics, genetically modified organisms, and regenerative medicine to evolve rapidly in this competent research field [4]. Following extensive attempts, researchers finally relied on a clue to manipulate the genome, which is inducing “DNA damage”. This is a life-threatening phenomenon where the most well-known detrimental type of DNA damage is the formation of double-strand breaks (DSBs). Exogenous DNA-damaging agents include ionizing radiation (IR), chemicals, pollutants, and some anti-tumor drugs. In contrast, endogenous DNA harmful agents have oxidative reactions with water where DNA is engaged, and reactive oxygen species (ROS) are the primary examples for generating DSBs, which, if unrepaired properly, might lead to genomic instability as well as the development of various types of cancer [5]. However, there are known “life-savers”, which are the DNA repair systems capable of fixing such breaks via one of two repair pathways: homology-directed repair (HDR) or non-homologous end joining (NHEJ). The cooperation of NHEJ and HDR repair pathways in the cell cycle occurs in a dependent manner where the NHEJ pathway is predominant in most cell cycle stages, being primarily active during the G1 phase, and as the name implies, it joins the broken ends of the double-stranded DNA without the requirement of a homologous sequence template. The HDR pathway instead requires a homologous sequence template and is predominant during the S/G2 phase where the sister chromatid is present [6,7,8]. It has been shown that the HDR pathway is the most dominant DSBs repair pathway in yeast Saccharomyces cerevisiae over the NHEJ pathway, while the opposite is true in higher eukaryotic species [9,10]. Furthermore, the NHEJ repair pathway is classified as error-prone because errors mainly occur upon processing the ends of the broken double-stranded DNA by either adding or removing nucleotides, generating frameshift mutations in DNA-transcribed regions that might be either mutagenic or lethal to the cells [11]. Contrary to the NHEJ repair pathway, the HDR pathway is classified as error-free because the repair of the DSBs does not rely on the addition or removal of nucleotides at the broken ends of the double-stranded DNA, but on the precise incorporation of new DNA fragments in homology with the neighboring lesion sequences based on homologous recombination [12]. Extensive research on DNA repair systems developed the concept of targeted DSBs at specific DNA sequence regions, a secret treasure lying behind the tremendous efficacy of genome editing strategies. The three most well-known classes of nucleases utilized as genome editing tools are transcription activator-like effector nucleases (TALENs), zinc finger nucleases (ZFNs), and clustered regularly interspaced palindromic repeats (CRISPR) and its associated nucleases (Cas) [13]. This review article will discuss the genome-editing tools CRISPR-Cas9, ZFNs, and TALENs. It will also encompass the importance of CRISPR-Cas9 technology in generating cellular disease models for novel therapeutics, its applications in gene therapy, and challenges with possible solutions for enhanced specificity.

## 2. Genome Editing Tools

The toolbox of genetic modification is constantly evolving to provide us with more precise, wider-scale, and diverse implementations, allowing us to gain innovative insights into the biological field [14]. Previously, genetic engineering was fulfilled by homologous recombination (HR), which involves the exchange of genetic material between two homologous double-stranded DNA sequences. Still, another alternative was required due to its low efficacy in many cell types and animals [15]. Targeted genome editing has been demonstrated to be a practical approach in biological research [16]. Since the discovery of restriction enzymes, the ability to effectively and accurately modify the genomic DNA sequences of cells harvested from humans, animals, and plants has become a significant subject in the research area. Nucleases applicable for genome editing can be engineered to specifically recognize sequences of interest and modify them, leading to the formation of DSBs, which can be either repaired by NHEJ or by HDR depending on the presence of a homologous DNA template [17]. This part will review the three main genome-editing tools: TALENs, ZFNs, and CRISPR-Cas9 (Figure 1).

### 2.1. ZFNs

Many types of zinc fingers exist. The predominant class of transcription regulators in the human genome are proteins harboring Cys2His2 zinc fingers. Some can bind to RNA and others mediate protein–protein interactions, but the majority are critical interactors with DNA as a matter of specific binding with its target. The most well-known zinc finger module utilized for genome editing comprises around 30 amino acids that join a zinc ion with two histidines and two cysteines to maintain this module’s stability [18]. These zinc fingers, occurring as repeated sequences of 30 amino acids, were first unraveled by Aaron Klug and his team in the transcription factor TFIIIA of *Xenopus laevis* [19,20]. The 3D characterization of the zinc finger’s structure indicated a specific code where consecutive zinc fingers via their specific amino acid side chains can bind to a consecutive triplet of nucleotides at three key sites within the target DNA region [21]. Acknowledging this potential, various teams, including Berg’s lab, began constructing new zinc fingers for more possibility of triplet nucleotide recognition [22]. To establish genome editing using zinc finger modules, two start-up companies were developed. The first one was Sangamo Biosciences, developed in 1995 to develop synthetic zinc finger transcription factors. The second one was Gendaq Ltd., developed in 1999 and aimed to generate zinc finger libraries with a broad accuracy scale. Zinc finger nucleases (ZFNs) can be defined as hybrid proteins composed of two major parts a nonspecific cleavage domain isolated from the bacterial Type IIS endonuclease restriction enzyme *Fok*I, which was discovered in Flavobacterium okeanokoites and a specific DNA binding domain which is composed of repeated Cys2His2 zinc fingers [23]. *Fok*I restriction endonuclease is naturally composed of both binding and cleavage domains where it can recognize a sequence of five base pairs (5′-GGATG-3′) and can chop 9 and 13 bases aside on the two DNA strands, indicating the absence of specificity at the target cutting site. Srinivasan Chandrasegaran’s team demonstrated that the two domains of *Fok*I restriction endonuclease could be dissociated from each other, and the cleavage domain of *Fok*I could be reengineered with a specific DNA-binding domain of interest depending on the target region of the DNA to be targeted [24]. The majority of the restriction enzymes exist in homodimers or homotetramers and cleave target sites in a symmetrical manner. It was quite mysterious how *Fok*I cuts both DNA strands, and it was occurring as a monomer in solution. Finally, monomers have been shown to actually dimerize upon binding to the target DNA. This feature allowed *Fok*I endonuclease to gain more advantages over other restriction enzymes because a monomer of the *Fok*I cleavage domain will not cleave the target DNA until it dimerizes. For genome editing to succeed, two ZFNs must bind specifically to the target DNA via the Cys2His2 zinc fingers DNA binding domain so that fused *Fok*I cleavage domains come into close proximity, dimerize, and undergo nonspecific cleavage of the target DNA [25,26,27]. Using ZFNs genome editing, the yellow gene of Drosophila melanogaster (fruit fly) was the first to be edited [28,29]. Switching to humans, ZFNs were quite effective in modifying interleukin-2 receptor subunit gamma (IL2RG), noting that patients with X-linked severe combined immunodeficiency (X-SCID) have a defective *IL2RG* gene [30]. This effectiveness paved the way for ZFNs to be extended in gene therapy applications. By 2010, ZFNs were successfully utilized for genome editing in various organisms, including flies [28,31,32], nematodes [33], zebrafish [34,35,36], sea urchins [37], mice [38,39], rats [40,41], and humans [30,42,43,44,45]. In terms of medical applications, genome editing using ZFNs entered phase I clinical trials to knockout the C-C chemokine receptor type 5 (*CCR5*) gene, a co-receptor expressed on CD4+ T lymphocytes, aiding the entry of human immunodeficiency virus to develop acquired immunodeficiency syndrome (AIDS). Knocking out this gene will allow CD4+ T lymphocyte cells to be resistant to the HIV virus, thus inhibiting its progression in the human body [46]. It is worth mentioning that due to its tremendous outcomes, scientists are also planning to knock out earlier hematopoietic progenitors. Despite its advantages, ZFNs were quite expensive and hard to handle in laboratories, so a better candidate was required as a simple gene-editing tool (Figure 2).

### 2.2. TALENs

Some bacteria of the genus Xanthomonas are classified as plant pathogens, negatively impacting crop yields [47]. These pathogenic bacteria release a cocktail of protein effectors to introduce the infection to the plant host cells, including transcription activator-like effectors (TALEs). These proteins act as virulence factors and play a key role as transcriptional activators, which bind to specific promoter regions via their tandem repeats upon entering the host cell’s nucleus via importin-α encoding for genes responsible for promoting the infection mechanisms [48]. TALEs are well known for their unique DNA-binding domain in their central region, ranging between 7 and 34 homologous repeated modules. Each module comprises around 34 amino acids that appear to be highly conservative except for two amino acids at the twelfth and thirteenth position, which are polymorphic and constitute the repeat-variable diresidue (RVD) and appear to be key regions to unravel the target nucleotide, which will be bound by the repeated module reflecting the high specificity and accuracy of TALEs [49]. Each repeated module comprises two alpha-helices linked by a short “RVD loop” forming the loop helix secondary structure [50]. TALENs are better than ZFNs in terms of simplicity. They can only bind to one nucleotide instead of three nucleotides, implementing an increased specificity with a decrease in the off-target events. In terms of genome editing, merging the unique specific DNA-binding domain of TALEs with a catalytic or functional domain of a nonspecific *Fok*I endonuclease generates the transcription activator-like effector nucleases (TALENs), which can effectively induce the formation of DSBs to the target DNA of interest [51,52,53,54]. Tomas Cermak and other groups proved this fusion and the possible generation of DSBs in vivo. Nonspecific *Fok*I endonuclease must dimerize to exert its catalytic activity; this is achieved by binding the engineered TALEs on opposite sides of the target DNA, allowing the catalytic domains of *Fok*I to become nearby and dimerize, thereby generating these DSBs [55]. TALENs offer a very accurate genetic engineering with minimal toxicity [56]. This genome-editing tool was utilized for various cell lines, including induced pluripotent stem cells (iPSCs) [57,58], as well as for generating knockouts in rats [59], nematodes (Caenorhabditis elegans) [60], and zebrafish [61,62]. Recent studies also showed that TALENs could exhibit an antiviral activity by inhibiting the Hepatitis B virus (HBV) in monolayer cultures and in vivo [63,64]. The inactivation of Epstein–Barr viruses was also possible via the disruption of Epstein–Barr virus (EBV)-encoded nuclear antigen-1 (EBNA1), which plays a key role in the replication and persistence of EBV [65]. Furthermore, in medical applications, TALENs are being utilized to enhance the efficiency of chimeric antigen receptor CAR-T cells genetically engineered for immunotherapy applications [66]. In the context of gene therapy, Jim Hu and his team utilized TALENs for the genetic correction of the cystic fibrosis transmembrane conductance regulator (*CFTR*) gene, which was mutated in a human cell line, and succeeded in restoring its channel activity [67]. In addition to that, Fanyi Zeng and his team also used TALENs to treat a widely common blood disorder, β654-thalassaemia. The Hemoglobin Subunit Beta (*HBB*) gene is responsible for the arousal of β654-thalassaemia, so by the deletion of the *HBB* IVS2-654 site of mutation, the symptoms of this disease were alleviated and provided an opportunity for further enhancing the efficiency and performance in future studies [68]. Even though they are highly specific, TALENs have an obvious disadvantage, which is being quite large in size. A cDNA is typically around 3 kb for a TALEN encoding. This makes it difficult in principle to deliver and express a pair of TALENs into the cells of interest, and the TALEN’s size will be less attractive in the therapeutic era because they cannot be packaged in vectors of limited cargo size such as adeno-associated viruses (AAV) or be delivered as RNA molecules. Another novel candidate was required to simplify genome-editing applications.

### 2.3. CRISPR-Cas

CRISPR-Cas has revolutionized the field of genome editing. In 1987, Ishino and his research team initially found CRISPR arrays in the genome of *Escherichia coli* when a unique repeating DNA sequence was uncovered inadvertently during their study of genes implicated in phosphate metabolism [69]. Due to the lack of enough DNA sequence data, the function of CRISPR arrays remained elusive until the mid-2000s. Afterward, the arrays were also discovered in archaea for the first time in 1993, particularly in *Haloferax mediterranei* [70]. These clusters of repeated DNA sequences were present in two out of three major domains of life, indicating that they are crucial and have a significant role. Meanwhile, various genes previously thought to produce proteins involved in DNA repair unique to hyper thermophilic archaea were shown to be tightly linked to the CRISPR locus [71,72]. They were termed *Cas* (CRISPR-associated genes). After extensive research, it appeared that there is a sequence similarity between the spacer regions of CRISPR loci in bacteria and archaea with DNA sequences of bacteriophages and archaeal viruses, concluding that CRISPR-Cas added a novel phenomenon, which is having an adaptive immune system against invading viruses. Based on the sequencing results, this system is present in around 90% of archaeal genomes and 40% of bacterial genomes. Furthermore, it can be analogous or equivalent to the RNA interference (RNAi) response in eukaryotes [73,74]. CRISPR-Cas systems can be classified into two major classes, and each class can be subdivided into three types (type I, III, IV, and type II, V, VI, respectively) based on the type and number of Cas proteins involved. The locus consists of spacers that are unique non-repeated sequences (20–75 nt) originating from foreign bacteriophages or archaeal viruses flanked by conserved repeated DNA sequences (20–30 nt) [75]. Within class II, the type II system is studied chiefly due to its simplicity compared to other types and classes. The CRISPR-Cas9 system was discovered in *Streptococcus pyogenes* by Jennifer Doudna and Emmanuelle Charpentier. This adaptive immune system involves three significant steps: (a) adaptation, which starts upon the first entry of the foreign viral DNA into the host where it is recognized by Cas2 and Cas3 endonucleases that excise the viral DNA to generate a pre-spacer, which becomes integrated into the spacer region of the CRISPR locus, and then upon the second entry of the viral DNA, (b) expression of the CRISPR locus at the transcriptional level generates a long non-coding precursor-crRNA (pre-crRNA) that requires further maturation and processing events by specific RNases such as RNase III to generate small stretches of mature crRNA. Together with another non-coding RNA, the tracking RNA (tracRNA) will be involved in the recruitment of Cas9 endonuclease, stabilizing the Ribonucleoprotein complex. Then, (c) interference occurs upon the base-pairing of crRNA with the target viral DNA, where the Cas9 protein also binds to a protospacer adjacent motif (PAM) sequence, which comprises around 2–5 base pairs in length following the target DNA, and cleaves the target viral DNA (proto-spacer), leading to its degradation (Figure 3). The type II CRISPR system from Streptococcus pyogenes was afterwards utilized for genome editing by a research team led by Jennifer Doudna and Emmanuelle Charpentier in 2012 [76]. They created a single guide RNA by fusing the crRNA to the tracrRNA (sgRNA) via a linker loop. This sgRNA relies on the Watson–Crick base-pairing rule to recruit the Cas9 nuclease to specified genomic regions [77]. Targeted genome editing by the CRISPR-Cas9 system generates DSBs, which are repaired by either NHEJ or HDR. Cas9 nuclease exhibits a bilobed design constituting a variable alpha-helical lobe acting as a structural domain for the binding to the double-stranded DNA, and a nuclease lobe composed of two nuclease domains: RuvC and HNH juxtaposed [78]. Emmanuelle Charpentier, Jennifer Doudna, and their research team proved, via experimental approaches, that each Cas9 nuclease domain cleaves a single DNA strand so that RuvC cleaves the non-complementary target DNA strand and HNH cleaves the complementary target DNA strand base pairing with sgRNA [76]. In genetic research, the Cas9-driven gene-editing method has been utilized widely to study the function of particular genes of interest to model several diseases and to demonstrate novel therapies in several models of genetic disorders. Cas9 platforms have been highly utilized in recent years due to their tremendous efficacy, especially from fungi [79] and plants [80,81] to various mammals [82,83,84,85], in generating transgenic organisms. This technique also simplifies the production of disease patterns for hereditary disorders and diseases such as cancer that permits a better understanding of the molecular underpinnings of these pathological processes [16,86,87]. By introducing numerous sgRNAs at the same time, Cas9 may simply be programmed for editing various genomic loci. This may be used to produce chromosomal rearrangements of vast dimensions. For example, establishing two DSBs in close proximity within the same chromosome can result in targeted deletions or inversions of DNA [88,89,90,91], and introducing two DSBs at two different chromosomes might induce targeted chromosomal translocations [92]. These target-mediated rearrangements generated by Cas9 can help produce patterns of diseases by modeling rearrangements in states of human disease (i.e., malignancies and inherited genetic diseases) [91,93]. Moreover, the introduction of permanent genetic alteration in the whole cells of the organism is now possible with high efficacy and can be bypassed through the future generation or offspring by simply injecting the CRISPR-Cas9 machinery (sgRNA; HDR template and Cas9 either in a protein or messenger RNA version) to the early-stage embryo or zygote to be genetically engineered at the germline level [94]. Furthermore, CRISPR-Cas9-mediated zygote editing in human embryos has also been demonstrated by Liang et al. [95], sparking a debate among research scientists and the general public opposing this action [96,97]. The researchers modified hemoglobin beta (*HBB*), the gene that causes the blood disease β-thalassemia when it becomes non-functional, using tripronuclear human zygotes. Later on, various unwanted off-targets were present, which limited zygote editing in clinical applications. In vivo somatic CRISPR-Cas9 genome editing also succeeded in correcting various genes responsible for severe diseases [98,99]. Lin et al. have shown that CRISPR-Cas9-mediated genome editing may be utilized to treat hepatitis B virus (HBV) infection in an elegant research work [100]. Despite all of its tremendous and successful applications in the genome-editing field, we must guarantee that CRISPR-Cas9 is designed well enough to target specific sites and that any off-target alterations with detrimental consequences are eliminated.

## 3. Disease Modeling and Gene Therapy Using CRISPR-Cas9

The fast creation of alternative in vivo and in vitro disease models has been assisted by CRISPR-based genome engineering technologies. Among the new options are the following: (i) the direct injection of Cas9 mRNA and sgRNAs into single-cell embryos for genome editing (Figure 4). This method has been extensively utilized to create rat [101], mouse [102], and monkey [82] models, showing the full potential contribution of the CRISPR-Cas9 technology for efficient and effective gene editing in the rapid development of genetically engineered animals where one or more have been changed at the same time. As an alternative to germline-modified mutant strains, (ii) in vivo gene editing involves the delivery of the CRISPR-Cas9 machinery system into the desired cells of interest in their natural tissue niches (Figure 5). CRISPR-Cas9 components can be delivered as either polymers, liposomes, or peptides, or packed in viruses. The introduction of viral vectors, primarily adeno-associated viruses (AAVs), which are non-enveloped, single-stranded DNA viruses, taking advantage of their small size, and non-pathogenic behavior has enabled in vivo edition [103,104]. These viruses are becoming more popular for use as vectors for therapies [105]. Nearly 80% of the human population is seropositive for AAVs, and to date, they are not linked to any type of human disease. Because of their low cytotoxicity, immunogenicity, and the very low chance of integration into the human genome, it is currently at the top of the list of leading in vivo delivery systems for the CRISPR components compared to other viral vectors [106,107]. Another key advantage is the presence of several well-known AAV serotypes for a specific and different tissue tropism, such as for heart cells, epithelial lung cells, neurons, and skeletal muscle cells, making them very efficient for an effective directed tissue-specific delivery for in vivo genome editing [108]. As mentioned previously, AAVs have a low risk of integrating into the human genome, which can be a barrier for novel therapy applications to a certain extent. Researchers ameliorated these vectors to generate a new variant termed the recombinant AAV (rAAV) vector, which can solve the problem of unintended genome integration since they are designed without the implication of Rep genes, which are involved in the modulation of viral gene expression, viral DNA replication, and the integration mechanism compared to AAV, which has these genes [109,110]. Currently in preclinical trials, the delivery of CRISPR-Cas9 machinery via the recombinant AAV vector has effectively shown an alleviation in atherosclerosis disease in mutant mice for low-density lipoprotein receptors (LDLR) [111]. Furthermore, the Food and Drug Administration (FDA) has approved the utilization of AAV vectors in hundreds of clinical trials [112]. The AAV delivery system has shown its potential to treat various in vivo disease models including muscular dystrophy [113,114], neurodegenerative diseases [115], sickle cell anemia [116], and metabolic liver diseases [117]. (iii) This is achieved using gene editing in conjunction with human induced pluripotent stem cells (iPSCs) to create models of genetically complicated diseases, given their potential of differentiation to any cell type of interest, if the appropriate cocktail mix is available and for future treatment strategies by correcting these disease models and reintroducing them to the patient. It is feasible to examine human genome alterations in a variety of genetic backgrounds using this method. In vitro disease pathogenesis modeling can be mimicked using iPSCs from patients that have been differentiated in culture to identify disease-affected cells. In iPSCs isolated from diseased people, CRISPR can be utilized to reverse specific mutations, explaining the consequences of such mutations and giving proof-of-principle for gene therapy (Figure 6). It seems inevitable that CRISPR-Cas9 technology will find its way into therapeutic applications, given the intense interest. This technology has been successfully deployed with in vitro and in vivo disease models, as well as to repair genetic abnormalities, according to several recent proof-of-principle studies (Figure 7).

### 3.1. CRISPR-Cas9 and Cancer Disease Modeling

A variety of genetic mutations in tumor suppressor genes and oncogenes are involved in developing various types of cancer. Designing a good cancer model is essential for studying cancer initiation and development processes and evaluating potential anticancer drug candidates. Outstanding research has been conducted to obtain precise and particular data where both in vivo and in vitro cancer models have been developed to answer various biological questions in research [118]. CRISPR-Cas9 combined with mouse cancer models has shown promise in recent research. It was possible to create a mouse liver cancer model by hydrodynamically injecting Cas9 expression plasmids and sgRNAs designed to target the wild-type version of tumor suppressor genes p53 and Pten. These mutations were capable of inducing liver cancer in mice. Furthermore, CRISPR-Cas9 components were delivered to adult mice’s pancreas utilizing a transfection-based multiplex delivery method in a recent study by Maresch et al. to alter multiple gene network sets [119]. The authors were also able to mimic the complicated chromosomal rearrangements, which were hallmarks for pancreatic cancer development. Based on the functional analysis of candidate genes in cancer mice models, Sanchez-Rivera et al. [120] presented an interesting strategy to mimic lung cancers. As a proof-of-concept, they employed a lung cancer model driven by a K-RAS (G12D) alteration. They also succeeded in generating lung adenocarcinomas in mice by using CRISPR-Cas9 machinery to alter tumor suppressor genes that are well known to have a loss of function in human lung cancer. In addition to that, several kinds of genomic rearrangements associated with lung cancer, including the EML4-ALK, KIF5B-RET, and CD74-ROS1 gene fusions, have been successfully induced using CRISPR-Cas9 technology [91]. It may not be sufficient to alter one gene at a time to create a complex polygenic tumor model. The co-delivery of several sgRNAs allows CRISPR-Cas9 to edit several targets at once. Because of this, it is a flexible tool for creating a human-like tumor model. A colorectal cancer model was created using the CRISPR-Cas9 genome-editing technology developed by Matano et al. by introducing multiple driver gene mutations [121]. Colon cancer models in the form of organoid structures were constructed in vitro using CRISPR-Cas9 by causing tumor-suppressing gene (TP453, APC, SMAD4, etc.) mutations and disrupting oncogenes (PI3K, KRAS, etc.). Roper et al. also introduced CRISPR mouse tumor organoids with a colonoscopy via mucous injections, delivering CRISPR-Cas9 components to the distal colon of the mouse through viral vectors [122]. Recent research was also capable of successfully generating a leukemia model by restoring many inactivated oncogenes via the delivery of the Cas9-sgRNA system lentivirally into the primary hematopoietic stem and progenitor cells (HSPCs) [123]. Using the CRISPR-Cas9 technology, Heckl et al. generated a mouse model of acute myeloid leukemia by inducing numerous mutations in epigenetic modifiers, transcription factors, and cytokine signaling genes. The cluster in this research targeted several genes including Dnmt3a, Nf1, Smc3, Ezh2, Tet2, and Runx1, which were all pooled together. HSPCs were able to generate myeloid neoplasia. Mouse brain tumors (glioblastoma and medulloblastoma) developed after the deletion of a single gene (Ptch1) or multiple genes (Trp53, Pten, and Nf1) by Zuckermann et al. Additional models using rats, pigs, and monkeys can also give better insights for pharmacological research and disease modeling [124]. Developing tumor organoid models was also possible using CRISPR-Cas9 technology [125,126]. In the future, using CRISPR-Cas9 technology to develop accurate cancer models will substantially enhance the investigation of functional cancer genomes and promote cancer treatments.

### 3.2. CRISPR-Cas9 and Cardiovascular Disease Modeling

Cardiovascular disease is a significant health threat and is the most common leading cause of mortality in many developed nations. A single genetic mutation or a mix of uncommon inherited heterozygous mutations is generally linked to several distinct kinds of cardiovascular disease [127]. Clinical therapies now focus on relieving illness symptoms rather than addressing any hereditary abnormalities. The capacity to evaluate the ability of gene therapy to control particular gene expression and enhance gene functions has been made possible thanks to the development of in vivo cardiovascular disease models using gene-editing technology and the in-depth research of harmful cardiovascular disease genes and their molecular processes. CRISPR-Cas9 has the potential to be used to generate cardiovascular disease models. An alpha myosin heavy chain protein is encoded by the myosin heavy chain 6 (MYH6) gene. Cardiomyocytes express this protein, which is part of a more prominent protein known as type II myosin. According to a recent study demonstrated by Carroll et al., Myh6 sgRNA was delivered to cardiac-specific Cas9 transgenic mice to generate mouse cardiomyopathy and heart failure models by disrupting the Myh6 locus [128]. Adenovirus-delivered Cas9 and sgRNAs targeting proprotein convertase subtilisin/kexin type 9 (PCSK9) were used by Ding et al. [99] in mouse liver in 2014 to induce targeted loss-of-function mutations into this endogenous gene. PCSK9 plays a crucial role in regulating LDL receptors. PCSK9 attaches to the LDL receptor and is internalized where lysosomal degradation occurs, and the LDL receptor is not recirculated to the hepatocyte surface under normal circumstances. LDL uptake is decreased due to PCSK9 inhibiting hepatic LDL receptor expression, leading to increased LDL cholesterol levels in the blood. If not maintained under normal levels, this might be a key driver for cardiovascular diseases [129]. The scientists recorded a significant incidence of mutation (about 50%) in PCSK9, who also looked at their pathophysiological consequences. A drop in cholesterol levels was seen. The authors noticed a decrease in the blood cholesterol levels in mice after targeting PCSK. Abrahimi et al. utilized CRISPR-Cas9 technology to introduce double-gene knockout to ablate the major histocompatibility complex class II (MHCII) in normal human endothelial cells [130]. These cells can generate vascular structures without inducing the activation of allogeneic CD4+ T lymphocytes. The use of such technology in allograft bioengineering, particularly the refining of heart transplantation, looks promising. Furthermore, CRISPR-Cas9 technology can precisely delete CCR5 and β2M from CD34+ HSCs while preserving their potential to multi-differentiate, paving the way for the future therapy of ischemic heart disease with HSCs. In combination with CRISPR-Cas9 technology, human induced pluripotent stem cells (iPSCs) can also provide an in vitro congenital heart disease model linked with GATA4 mutations to investigate the pathophysiology of this gene mutation [131]. Wang and his colleagues showed that the TAZ gene mutation is correlated with defects in the myocardial structure, metabolism, and function using Barth syndrome (BTHS) iPSC-derived cardiomyocytes (iPSC-CMs) and genome editing [132].

### 3.3. CRISPR-Cas9 and Neurological Disease Modeling

Neurodegenerative diseases (ND) are a set of disorders that have garnered the most significant concern because of the lack of precise diagnostic techniques or therapies, which include Huntington’s disease (HD), Alzheimer’s disease (AD), and Parkinson’s disease (PD) [133,134]. Several possible pathogenic processes underlying NDs include protein build-up with aberrant structures [135], oxidative stress [136], and ubiquitin–proteasome and/or autophagic lysosomal tracts [137]. The generation of neurological disease models has been described in several high-quality research papers. To better understand HD’s etiology, Yan et al. utilized CRISPR-Cas9 to create a genome-edited HD pig model that expressed full-length mutant Huntington (HTT) internally [138]. The creation of the HTT gene knock-in swine would be a promising breakthrough in Huntington’s ND research and treatment exploration. HTT would also be of major importance for pathogenesis study. Using CRISPR technology in vivo, Tabebordbar et al. edited the genome of the mouse muscle as well as stem cells carrying frameshift mutations in the Duchenne muscular dystrophy (*Dmd*) gene [139]. Both ends of Exon 23 in the mutated *Dmd* gene were targeted by Cas9 endonuclease and sgRNAs, being delivered by AAVs, and were successfully able to generate a truncated, but functional, protein. Muscle functional deficits were partially addressed within this therapy. The authors claimed that this kind of treatment was able to partially resolve muscle functional deficiencies. Liu et al. investigated the underlying mechanism of epilepsy produced by SCN1A loss-of-function mutations using CRISPR-Cas9-mediated genome editing in an iPSC-based model [140]. Paquet et al. utilized hPSCs, modified them with a CRISPR-Cas9-based genome-editing system, and successfully generated a cellular model with Alzheimer’s disease-causing mutations in amyloid precursor protein and derived cortical neurons were produced, which demonstrated genotype-dependent disease-associated behaviors [141]. Leucine-rich repeat kinase 2 (LRRK2) and alpha-synuclein (SNCA) disruptions often lead to the autosomal dominant Parkinson’s disease, whereas phosphatase and tensin homolog-induced kinase 1 (PINK1), parkin, ATPase type 13A2 (ATP13A2), and DJ-1 disruptions often lead to the autosomal recessive Parkinson’s disease [142]. Soldner et al. integrated CRISPR-Cas9 genome editing with genome-wide epigenetic information to provide a genetically precisely controlled experimental system in human iPSCs. In non-coding distal enhancer regions regulating SNCA expression, this approach has found PD-associated risk mutations; it has also established that SNCA transcriptional disruption is connected to sequence-dependent binding of the brain transcription factors NKX6-1 and EMX2 [143]. These results show that the genetic modification approaches can produce unique ND animal models for future research.

### 3.4. CRISPR-Cas9 and Hereditary Eye Disease Modeling

The genetic diagnosis of several hereditary eye diseases such as retinitis pigmentosa (RP), leber congenital amaurosis (LCA), congenital cataracts, retinoblastoma (RB), congenital glaucoma, congenital corneal dystrophy, congenital corneal dystrophy, and Usher syndrome has been made more explicit over recent years through the promotion of genetic sequencing technology [144]. Retinitis pigmentosa (PP) can be defined as a heterogeneous group of retinal inherited diseases leading to the bilateral degradation of rod–cone photoreceptors and visual sight loss. In the context of animal RP models, CRISPR-Cas9 has previously been employed. The shape of the endoplasmic reticulum is influenced by the receptor expression-enhancing protein 6 (REEP6), which is a member of the REEP/Yop 1 family of proteins and is required for the retinal photoreceptors’ viability and their functionality [145]. Arno et al. demonstrated that the autosomal-recessive retinitis pigmentosa is due to biallelic mutations in REEP6 [146]. In individuals with RP from unrelated families, variations of REEP6 were found. Furthermore, a knock-in REEP6 p.Leu135Pro mouse model was generated via CRISPR-Cas. In homozygous knock-in mice with REEP6 p.Leu135Pro, the clinical symptoms were similar to retinitis pigmentosa, such as photoreceptor degeneration and rod photoreceptor malfunction. This gives a better model for animals for future RP investigations. The rodless (rd1) mouse, the most widely used preclinical RP model, has been the subject of intense discussions almost a century after its onset, as its etiology remains uncertain. In chromosome 5 of the Pde6b site, the rd1 mouse possesses two homozygous variant mutations: a murine viral leukemia virus (Xmv-28) insertion site oriented in the opposite direction of intron 1, and a nonsense point mutation (Y347X) [147,148]. CRISPR-Cas9 has been utilized by Wu et al. to rescue and alleviate the illness, and the results showed that the Y347X nonsense point mutation has a pathogenic effect in rd1 [149]. Combining CRISPR-Cas9 technology with other strategies opens up new options for treating associated eye disorders, such as iPSC and AAV therapy. Bassuk et al. were the first to report that CRISPR/Cas9 precisely repairs retinitis pigmentosa GTPase regulator (RPGF) point mutations that cause X-linked RP in patient-specific iPSCs; this supports the notion that combining gene editing with autologous iPSCs could be a personalized iPSC transplantation strategy for therapies of various retinal degenerations [150]. LCA is a retinal disease that causes sight loss at an early stage of life [151]. To test whether a mutation in human Potassium Inwardly Rectifying Channel Subfamily J Member 13 (KCNJ13) causes LCA, Zhong et al. generated mice models with the Kcnj13 mutation by injecting spCas9 mRNA and sgRNA into the zygote. Mice models with the Kcnj13 mutation showed a reduced light response, aberrant rhodopsin localization, and loss of light receptors in the retina, concluding that the human LCA disease clinical outcomes can be mimicked using mice models with the loss of the Kcnj13 function [152]. RB is the most common eye cancer in children’s developing retinas. Homozygous retinoblastoma 1 (RB1) mutations cause RB cases [153]. Naert et al. developed a fast preclinical model of RB by simply inducing an RB1 knockout via the CRISPR-Cas9 system for the induction of RB1 and retinoblastoma-like 1 (RBL1) knockout *Xenopus tropicalis* [154]. Animal models have developed RB within a short period of time, appearing to be a good candidate model for drug discovery and the rapid identification of novel therapeutic targets. Jian Tu et al. used the CRISPR-Cas9 nickase system to create an RB1 heterozygous deletion in pluripotent H1 human embryonic stem cell line, which provides an exciting cell resource for further investigation of the etiology of retinoblastoma [155]. Glaucoma, the world’s second leading cause of sight loss, is well known to develop an increased intraocular pressure (IOP) [156]. Myocilin protein is encoded by the *MYOC* gene. The latter is expressed in ocular tissues and plays a vital role at the cytoskeleton level. Myocilin (MYOC) mutations have been linked to primary open-angle glaucoma (POAG) [157,158,159]. When mutant myocilin accumulates inside cells, it induces the activation of the unfolded protein response (UPR) cascade and causes stress to the endoplasmic reticulum (ER), which is found in the trabecular meshwork (TM) [160,161]. Chronic ER stress causes TM cells to die, resulting in an augmented IOP and glaucoma [162,163]. Jain et al. used CRISPR-Cas9 technology to cut down the expression of the mutant version of MYOC in a POAG mouse model. The outcomes showed reduced ER stress, decreased IOP, and the avoidance of additional glaucomatous damage in the mouse’s eyes. They also established the potential of using CRISPR-Cas9 in human eyes with glaucoma for future therapeutic treatments [164].

### 3.5. CRISPR-Cas9 and Infectious Disease Modeling

The CRISPR-Cas9 system was first found as an antiviral adaptive immune system in bacteria, but it can also combat viral infections in humans. A study has shown that the transient transfection of cells transduced with a GFP reporter lentivirus with CRISPR-Cas9 components resulted in the targeted destruction of both pre-integrating HIV-1 viral genomes and integrated HIV-1 proviruses [165]. Using CRISPR-Cas9 technology in vitro, Ebina et al. mutated a long terminal repeat (LTR) region in human immunodeficiency virus 1 proviral DNA (HIV-1). Furthermore, latently infected T lymphocyte cell lines, macrophages, and monocytes by HIV-1 were utilized as targets for CRISPR-Cas9, providing long-term resistance against this viral infection [166]. HIV-1 infects the host cells primarily through C-C Motif Chemokine Receptor 5 (CCR5). A CCR5-Δ32 mutation either slows down the virus’s progression in HIV-infected individuals or halts the HIV infection by acquiring resistance to the HIV infection. Ye et al. were able to mimic this phenomenon by generating iPSCs, homozygous for the CCR5-Δ32 mutation, by genome editing via the combination of either CRISPR-Cas9 or TALENs with the piggy BAC technology [167]. Globally, chronic hepatitis caused by the hepatitis B virus (HBV) is one of the most frequent infectious disorders. CccDNAs (covalently closed circular DNA) of HBV, present inside infected cells, are difficult to remove with existing treatment options. Researchers found that CRISPR-Cas9 designed to target HBV cccDNAs, when introduced into cells, can successfully target the HBV genome and help in its clearance in several in vitro and in vivo approaches [168,169]. Ramanan et al. showed that the inhibition of chronic HBV gene expression and replication can be achieved by the CRISPR-Cas9 genome-engineering tool targeting and cutting conserved areas of the chronic HBV genome [170].

### 3.6. CRISPR-Cas9 and Immunodeficiency Disease Modeling

Multiplex CRISPR may be used to create various immune-deficient mouse strains, as Huang and his team demonstrated in 2014 [171]. The authors targeted Mouse Beta-2-Microglobulin (B2m), Interleukin 2 Receptor Subunit Gamma (Il2rg), Perforin 1 (Prf1), Protein Kinase, DNA-Activated Catalytic Subunit (Prkdc) and Recombination Activating 1 (Rag1) genes by injecting Cas9 mRNA and numerous sgRNAs into the embryos of the mouse strains. Hashikawa and his team also succeeded in generating a stable strain of rabbits with X-linked severe combined immunodeficiency (X-SCID), harboring a phenotype that encompasses hyperplasia of the thymus and the loss of B and T cells by targeting the *Il2rg* gene via the CRISPR-Cas9 system [172]. Ren and his colleagues also developed Bama IL2RG^−/Y^ pigs via CRISPR-Cas9 technology and aimed to use their immunodeficient model as a host for human cancers to pave the way for preclinical models for anti-cancer treatments [173].

## 4. CRISPR-Cas9 and Gene Therapy

The manipulation of DNA or RNA for the treatment or prevention of human diseases is known as gene therapy. By disrupting endogenous disease-causing genes, correcting disease-causing mutations, or introducing new protective genes, precise genome editing holds the potential to permanently cure a wide variety of life-threatening diseases. Various clinical trials were launched to treat blood disorders [174], cancers [175,176,177], and other currently ongoing trials to treat eye diseases, chronic infections, and rare protein-folding diseases. For gene therapy applications, CRISPR-Cas9 provides a unique, highly effective genome-editing technique.

### 4.1. Monogenic Diseases Correction by CRISPR-Cas9

When a single gene in the human DNA is defective, it is classified as a monogenic disorder. Gene treatments are more effective in treating monogenic diseases than polygenic diseases, such as cancer. In the context of monogenic disorder treatment, CRISPR-Cas9-based gene therapy offers significant advances over other technology due to its simplicity and versatility. In CRISPR-Cas9 gene therapy, the repair of monogenic diseases is currently the most translatable application. This strategy to correct a disease-causing mutation in the mouse embryo was initially utilized by Wu and others [178]. Their research relied on treating a dominant loss of function mutation occurring in the Crystallin Gamma C (*Crygc*) gene, which causes severe symptoms such as eye clouding that leads to reduced eye vision, termed a cataract. This research team introduced the CRISPR-Cas9 machinery (Cas9 in the messenger RNA format and a sgRNA designed to target the dominant mutant *Crygc* gene) by injecting it into the zygote. They succeeded in correcting this mutation by utilizing the wild-type allele of this gene as a template on the homologous chromosome. The mice were successfully able to pass on the corrected Crygc allele to their offspring. The same group also aimed to repair the mutant Crygc by CRISPR-Cas9 in mouse spermatogonial stem cells (SSCs) in a recent paper [179]. They demonstrated elevated levels of rescue efficiency with no indication of off-target consequences. Schwank et al. successfully edited the *CFTR* gene in human stem cells to repair a genetic disorder linked to cystic fibrosis using CRISPR-Cas9 [180]. Another team also utilized CRISPR-Cas9 machinery based on injecting Cas9 in the mRNA format, with sgRNA and exogenous single-stranded DNA oligos acting as HDR templates to replace the mutated dystrophin gene in mouse embryos. The dystrophin gene is responsible for Duchenne muscular dystrophy (DMD), an X-linked disease characterized by the atrophy of the muscles. The authors obtained partially corrected mosaic mice because they introduced the editing system after the zygote stage, but some skeletal muscle cells were able to achieve a complete rescue from this disease [181]. In vivo somatic CRISPR-Cas9 genome editing also succeeded in correcting multiple genes responsible for severe diseases. One of the first studies was launched by Yin et al., utilizing a mouse model for type I tyrosinemia. The buildup of cytotoxic metabolites and liver cell death is due to type I tyrosinemia disease caused by a deficiency in the fumarylacetoacetate hydrolase (FAH) enzyme. The authors succeeded in correcting the mutant fumarylacetoacetate hydrolase allele and stabilized the expression of this enzyme, thus rescuing the mice from losing excessive weight and liver cell death via injecting the CRISPR-Cas9 machinery (DNA oligos HDR and vectors encoding for Cas9 and sgRNA) into the mice tail vein, reaching the liver directly [98].

### 4.2. Non-Monogenic Disease Correction by CRISPR-Cas9

A flexible gene-editing technique such as CRISPR-Cas9 can be used to delete or insert genes, activate or repress genes, and alter epigenetics. Treatment of non-monogenic cancers, cardiovascular diseases, metabolic disorders, and Alzheimer’s disease may be possible with this technology. The interruption of the gene encoding the proprotein convertase subtilisin/kexin type 9 (PCSK9) in the mouse liver was the subject of an in vivo research study. PCSK9 is a hepatocyte-secreted protein into the plasma that acts as an LDL receptor antagonist, limiting the absorption and degradation of low-density lipoprotein (LDL) cholesterol. As a result, naturally occurring PCSK9 loss-of-function mutations lower blood cholesterol levels. The authors utilized adenoviral CRISPR-Cas9 vectors to alter PCSK9 in the mouse liver. This resulted in lower amounts of PCSK9 protein, higher levels of hepatic LDL receptors, and lower plasma cholesterol levels. It is worth mentioning that this method relies on NHEJ to impair a *PCSK9* gene’s function. It was feasible to achieve editing efficiencies of up to 50%, promising for clinical trials [102]. An AAV-based approach was used to target PCSK9 in the mouse liver by Ran et al., where this team also generated and utilized smaller versions of Cas9 orthologs isolated from Staphylococcus aureus (SaCas9) in this approach [182]. CRISPR-Cas9 technology has also been classified as a good candidate for cancer treatment. Using CRISPR-Cas9-mediated genome editing, Antal et al. corrected a loss-of-function Protein kinase C (PKC) mutation in a colon cancer cell line derived from a patient, leading to decreased tumor development in a xenograft model [183]. In research by Liu et al., CRISPR-Cas9-mediated bladder cancer cell-specific genome editing was achieved utilizing an AND logic gate. The term AND means that two inputs must be coupled to generate a single output. Both (two inputs) the urothelium-specific human uroplakin II promoter and cancer-specific human telomerase reverse transcriptase promoter and were used to regulate the production of sgRNAs and Cas9 in this work. Two promoters (two inputs) exist exclusively in bladder cancer cells, making it possible to produce both Cas9 and sgRNA only in bladder cancer cells. In this case, the sgRNAs were intended to target and disrupt the LacI gene, which under normal conditions is expressed as a normal protein that binds to the lac operon, suppressing the downstream effector gene expression (the output) acting as an inhibitor of oncosuppressers. LacI is targeted and mutated within the bladder cancer cells by sgRNAs specifically designed for LacI and Cas. The lac operon will be active, thus expressing downstream effector genes (oncosuppressors). As a result, E-cadherin, Bax (BCL2-associated X), and p21 expression are all classified as effector genes (the output), leading to migratory suppression, apoptosis, and cancer-specific growth inhibition, limiting tumor growth [184].

## 5. Challenges of CRISPR-Cas9 and Enhanced Specificity

Despite the enormous promise of CRISPR-Cas9 in genome editing. However, some major difficulties remain, such as off-target mutations, PAM dependency, strategies of delivery, the generation of genetically modified RNA (gRNA), and the host-undesired immune response against CRISPR-Cas9 machinery.

### 5.1. CRISPR-Cas9 Off-Targets

A key problem with CRISPR-Cas9-mediated genome editing is the occurrence of mutations that are not intended to be there. CRISPR-Cas9 has a higher risk of off-target mutations in human cells than TALENs and ZFNs [185]. Because CRISPR-Cas9 produces permanent genomic changes in gene therapy, its off-target consequences must be carefully assessed and managed. It is common for large genomes to have several DNA sequences that are similar or highly comparable to the DNA sequence that is the target. These similar DNA sequences are also cleaved by CRISPR-Cas9, resulting in mutations in unintended locations, termed off-target mutations. Cell transformation or death can arise from mutations that are not on-target. A huge effort in the research field is paid to eliminate the undesired off-target effects of CRISPR-Cas9 [185,186,187]. It is preferable to pick up target sites with the fewest off-target sites and mismatches occurring between gRNA and its corresponding target sequence to ensure CRISPR-Cas9 specificity. CasOT, a versatile searching bioinformatics tool created by Xiao et al., can unravel possible off-target locations throughout whole genomes [188]. Other factors causing off-target mutations include the dose of CRISPR-Cas9 utilized, which must be adjusted carefully [189]. DNA methylation appears to have no effect on CRISPR/specificity compared to TALENs [187]. Switching from wild-type Cas9 protein to Cas9 nickase can also assist in minimizing the number of off-target alterations while preserving the effectiveness of CRISPR-Cas9 on-target cleavage [186].

### 5.2. nCas9 and dCas9

There are two domains in the Cas9 endonuclease. The first domain is the large globular recognition domain (REC) linked to the small nuclease domain. Besides RuvC and HNH, the NUC domain has a PAM-interacting site [87]. When the guide RNA is loaded into Cas9, it undergoes a conformational reorganization to establish a central channel for RNA–DNA heteroduplex binding and recognition of conventional PAM motifs [190]. Crystallography studies have elucidated the mechanism of action of Cas9 endonuclease that formed a complex with duplexed target DNA with a PAM sequence and a guide RNA [191]. For genome editing, silencing, and transcriptional regulation, there are several Cas9 variants available today that have enhanced specificity and reduced off-target effects compared to the native Cas9. These variants may be used in diverse systems such as yeast, drosophila, bacteria, and humans.

#### 5.2.1. Cas9 Nickase

To enhance the specificity and reduce off-target events when using this technology, second-generation genome-editing tools were required. As previously mentioned, the RuvC nuclease domain cleaves the non-complementary target DNA strand and the HNH nuclease domain cleaves the complementary target DNA strand base pairing with sgRNA. Cong et al. created the first Cas9 nickase (Cas9n) by a mutation in the native Cas9 nuclease domain (RuvC-mutant D10A, aspartate to alanine substitution), which impairs the ability of Cas9 to cleave non-complementary target DNA strands, thus creating a single nick from the complementary side, which can be repaired faithfully by the mammalian cells without inducing DSBs and therefore limiting off-target scenarios (Figure 8) [186]. Interestingly, it is also possible to undergo another single-point mutation in HNH of native Cas9 (HNH-mutant H840A), which impairs the ability of Cas9 to cleave complementary target DNA strand, thus creating a single nick from the non-complementary side. A high-fidelity homology-directed repair method is generally used to repair individual nicks generated in the genome (HDR). To increase the specificity and reduce off-target events, Cas9n has been utilized in a pair of nickase systems with two distinct gRNAs that, when they are in close proximity and induce nicks at the same time, are capable of inducing DSBs that produce staggered ends instead of blunt ends and can be utilized for NHEJ or HDR [192,193,194]. Using the paired-nicking approach, HDR becomes more specific than the native Cas9 approach with a sharp decrease in the off-target events by 50- to 1500-fold less without altering the on-target cleavage efficiency [193]. Nickases can be useful tools for prime and base editors, which have the potential to edit the DNA sequence in a precise fashion without the need to introduce double-stranded breaks or donor DNA templates, thus having no reliance on the HDR pathway. Nowadays, base editors comprise a CRISPR-Cas9 nickase (incapable of forming DSBs due to a catalytic domain impairment) linked/fused to either proteins capable of triggering DNA repair or a single-stranded DNA deaminase enzyme. Base editors can currently be classified into two major classes: adenine base editors (ABEs), which have the capacity to convert A•T base pairs to G•C base pairs, and cytosine base editors (CBEs), which have the capacity to convert C•G base pairs to T•A base pairs [195,196]. Nearly 30% of human disease variants may be due to the four transition mutations (G→A, A→G, T→C, and C→T) [197]. Both base editors (CBEs and ABEs) can revert or install transition mutations and this was feasible in various cellular models and animals harboring human genetic diseases [198,199,200]. For increased precision, prime editing was introduced recently as an effective genome-editing technology that is capable of performing the whole 12 possible types of point mutations (insertions, deletions, and substitutions), as well as small regional insertions and deletions in a targeted and precise manner in the DNA of interest. Prime editors are composed of a CRISPR-Cas9 nickase domain linked/fused to a reverse transcriptase functional domain. This system also requires a uniquely engineered prime editing guide RNA (pegRNA), which has a spacer sequence that can guide the primer editor to the genomic DNA to be precisely edited. Furthermore, it has the designed edit of interest to be added in an extension region at the 3′ end following the nicking (single-stranded break) of the target DNA by Cas9 nickase. The prime editor acquiring the reverse transcriptase functional domain utilizes the newly released 3′ end of the target DNA region to prime/initiate reverse transcription using the extension region, which is present in the pegRNA as a template. Following reverse transcription, the newly edited DNA target site will exist as a 3′ DNA flap, which is redundant with a 5′ DNA flap having the unedited DNA sequence. Under normal conditions, DNA repair systems tend to remove the 5′ DNA flap, enhancing the incorporation of the edited 3′ DNA flap into the DNA target site, thus generating a heteroduplex DNA where one strand is edited and the other strand remains unedited. The edit is installed permanently with the help of a simple sgRNA to further direct the primer editor to induce another nick to the non-edited strand, thus allowing the DNA repair systems to faithfully use the edited strand as a template to re-synthesize the non-edited strand, thus generating a permanent edited DNA duplex. Several human cell lines [201], human induced pluripotent stem cells [202], mouse cortical neurons [202], and mouse embryos [203] were utilized for prime editing.

#### 5.2.2. Inactive dCas and Dimeric RNA-Guided FokI Nucleases (RFNs)

A nuclease-deficient mutant form of Cas9 that is catalytically inactive (dCas9) can be generated by inducing single-point mutations in the two-nuclease domains (RuvC and HNH) where this endonuclease can recognize the PAM sequence of the target DNA without cleaving [204,205]. Because of this modification, CRISPR interference (CRISPRi) and CRISPR activation (CRISPRa) may be performed with great efficiency and precision utilizing dCas9 with an effector and sgRNA. DCas9-sgRNA fusion with a repressor inhibits transcription initiation and elongation, in contrast to the notion of RNAi, which involves the destruction of transcripts or the inhibition of translation [206]. The fusion of dCas9-sgRNA with effectors such as VP64 can induce a positive regulation of genes at the transcriptional level (Figure 9) [204,205]. Thus, the dCas9-sgRNA system provides a unique platform for RNA-directed DNA targeting for an efficient and stable modulation of transcription. Dimeric RNA-guided *Fok*I nucleases (RFNs) have been created by joining the *Fok*I nuclease domain with an inactive dCas9 protein unable to cleave the target DNA (Figure 10). To achieve highly efficient genome editing, the dimerization of two RFNs is required. Because *Fok*I is well known to cleave the target just upon dimerization, this gives a plus point over the Cas9 nickase system in terms of high editing efficacy and significantly lower unintended off-target effects. The cleavage effectiveness solely depends on the binding of two gRNAs with specific spacing and orientation, which decreases the chances of the same target site appearing many times in the genome [207,208].

### 5.3. CRISPR-Cas9 Delivery Strategies

Cargo and delivery vehicles are the two main forms of delivery. Three methods are frequently mentioned when dealing with CRISPR-Cas9 cargos: (1) ribonucleoprotein complex (Cas9 protein with the single guide RNA), (2) a DNA plasmid that can express both the Cas9 protein and the single-guide RNA, and (3) Cas9 mRNA, which will be translated following delivery in the cytoplasm and afterwards will become coupled to the single-guide RNA [83,209]. Which of these three cargos can be packed and whether the system is useful in vitro or in vivo will frequently depend on the delivery vehicle that is being employed. Furthermore, the tight control of the optimal concentration to be considered for Cas9 and the guide RNA is necessary to prevent the risk of undesired cytotoxicity. For example, to introduce Cas9 as a DNA format, it becomes quite complicated to ascertain how many Cas9 units can be functional and perform the edit compared to the Cas9 protein. Delivery vehicles can be divided into three general categories: non-viral vectors, viral vectors, and physical vectors. The most well-known non-viral-vectors are cell-penetrating peptides, lipid nanoparticles, gold nanoparticles, and DNA nanoclews, whilst viral vectors can include lentiviruses and engineered AAVs. Electroporation and microinjection are also widely used as physical delivery approaches. It is worth adding that each approach or delivery method has its advantages and disadvantages, and parts of them can be quite specific to the type of delivery utilized. For example, the impact of delivering it to cells might be completely different from that when delivering it to a living organism. In addition, using ribonucleoprotein complex instead of mRNA or plasmid DNA will significantly decrease the unintended off-target events, which are critical for future therapeutic approaches. Moreover, the chosen delivery system must acquire two key characteristics: high specificity, which can effectively target the desired target of interest, and safety, because living organisms are always the concern and the priority. CRISPR-Cas9 should be given more attention to developing a new, robust delivery strategy [210,211].

### 5.4. CRISPR-Cas9 PAM Dependency

CRISPR-Cas9 technology may theoretically be applied to any target DNA sequence using a well-designed programmable gRNA. In addition to gRNA/target sequence complementarity, CRISPR-Cas9’s specificity relies on the presence of a 2–5 nt PAM region located directly downstream of the target sequence of interest [76]. Cas9 orthologs have various PAM sequences, such as NGG PAM from Streptococcus pyogenes [212], NNNNGATT PAM from Neisseria meningitides [213] and NNAGAAW and NGGNG from Streptococcus thermophiles [214,215,216]. Despite the fact that the PAM sequence increases the specificity of the CRISPR-Cas9 system, the requirement of this sequence also limits the frequency of finding targetable sites in the whole genome. You can discover a target site every 8 nt with NAG PAM and NGG PAM, but only every 32 and 256 nt with and NNAGAAW PAM and NGGNG PAM, which might be a limiting factor for genome editing. It is worth mentioning that the PAM dependency can be positively enhanced with the generation of Cas9 variants such as xCas9, which can recognize many PAM sites, including GAA, GAT, and NG in mammalian cells [217]. SpCas9-NG is designed to recognize NG sites [218] and the SpG variant can also easily target a wide set of NGN PAMs and can offer a versatile flexibility in genome editing. The SpG variant was further modified and optimized to generate a near-PAMless-SpCas9 variant termed SpRY, which can target almost all PAMs including NRN and, to a certain extent, NYN PAMs [219,220]. Hopefully, with the new Cas9 variants, robust activities can be achieved at a higher resolution level.

### 5.5. gRNA Synthesis and Modifications

For CRISPR/Cas9-mediated genome editing, gRNA production is another essential consideration. Unfortunately, RNA polymerase II cannot be used to produce gRNA because of the substantial post-transcriptional processing and mRNA modifications that occur after transcription. To generate gRNAs in vivo, RNA polymerase III, U3, and U6 snRNA promoters are currently utilized. U3- and U6-based gRNA synthesis is additionally limited by the absence of commercially accessible RNA polymerase III. Gao et al. developed an artificial gene, termed Ribozyme-gRNA-Ribozyme (RGR), which, upon expression, can generate the required sequences of the gRNA and ribozyme sequences at the side ends of the gRNA as an mRNA format [221]. After the self-processing and auto cleavage of the mRNA, mature desired gRNAs of interest were generated and were able to induce target sequence cleavage in vitro as well as in yeasts [221]. In addition, truncated guide RNAs (truRNA) are a simple and successful approach for improving the specificity of Cas9 nucleases or paired nickases by minimizing off-target effects. TruRNAs are variants of the original sgRNAs, shorter in length, comprising around 17 nucleotides. Because they have shorter target complementarity regions, shortened gRNAs have the potential to reduce unwanted mutations at some off-target locations without compromising the on-target genome editing efficiency.

### 5.6. Immune Response against Cas9 Endonuclease

The two major sources of Cas9 endonuclease are Streptococcus pyogenes and Staphylococcus aureus, which are well known for causing human diseases including strep throat and MRSA. Nearly 80% of individuals in the healthy population have shown to develop a cellular, T-cell mediated and a humoral, antibody mediated immunity against both bacteria. The human’s immune system acts as a protective shield against exogenous surface and secreted proteins. So, the question will highlight the fact that considering Cas9 is classified as an intracellular protein, what will be the future for in vivo Cas9-mediated gene editing for novel human disease therapies [222]?

## 6. Conclusions

To grab attention in the research era, genome-editing tools should be efficient, reproducible, easy to manipulate, inexpensive, and have the potential to target any sequence of interest in the genome without causing any unintended off-target effects. After addressing several obstacles, CRISPR-Cas9 technology can take the lead in this research battle because it encompasses all of the criteria required for an effective genome-editing tool. CRISPR-Cas9 can edit any cell or organism at a single-nucleotide resolution. The extensive usage of subsequent CRISPR-based applications has transformed it into a multifunctional platform with applications ranging from single-gene editing to multiplexed editing, sequence-specific control of gene expression, and genome-wide screening. These new breakthroughs have significantly expanded the technical options for unraveling novel gene functions and modeling various diseases in cells and organisms. More research is needed to investigate the unique characteristics and ameliorate the performance of CRISPR-Cas9 technology in terms of delivery strategies, specificity, and off-target effects.

## Figures and Tables

**Figure 1 cells-11-03615-f001:**
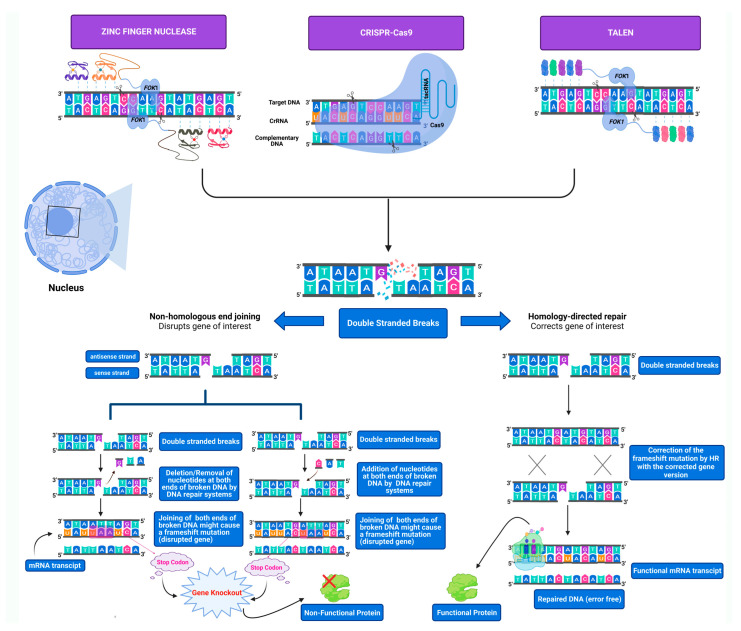
**Genome editing tools’ mechanism of action and the pathways involved in DSBs repair of targeted endogenous DNA.** DSBs are either repaired by NHEJ or HDR repair pathways. In the absence of a donor template, the NHEJ repair pathway will process the ends of the broken targeted double-stranded DNA by either inserting or deleting nucleotides, thus disrupting the open reading frame by generating frameshift mutations leading to the loss of the gene’s functionality. In the presence of a donor template, the HDR repair pathway will precisely incorporate the donor template DNA in homology with the neighbor lesion sequences based on homologous recombination. All three genome-editing tools (ZFNs, TALENs, and CRISPR-Cas9) generate DSBs that are either repaired by NHEJ or HDR depending on the aim of the research.

**Figure 2 cells-11-03615-f002:**
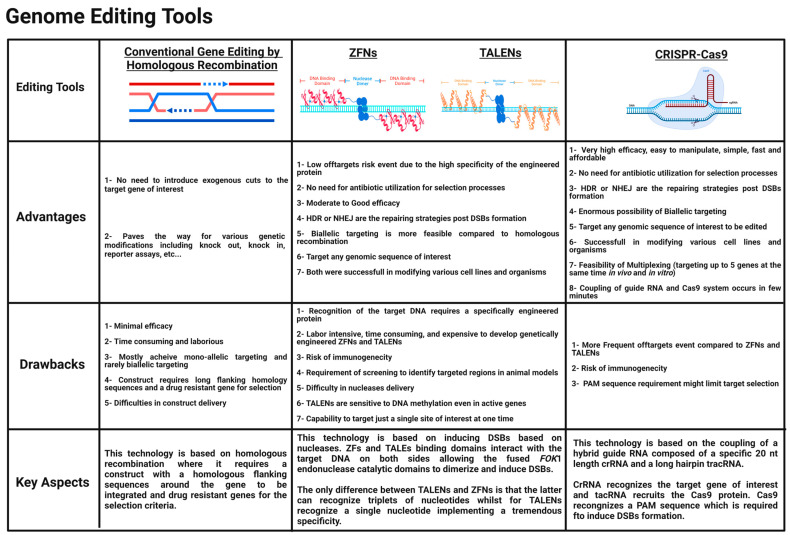
**Comparison of genome editing tools.** Comparison between homologous recombination (HR), ZFNs, TALENs, and CRISPR-Cas9 in terms of advantages, drawbacks, and key aspect features.

**Figure 3 cells-11-03615-f003:**
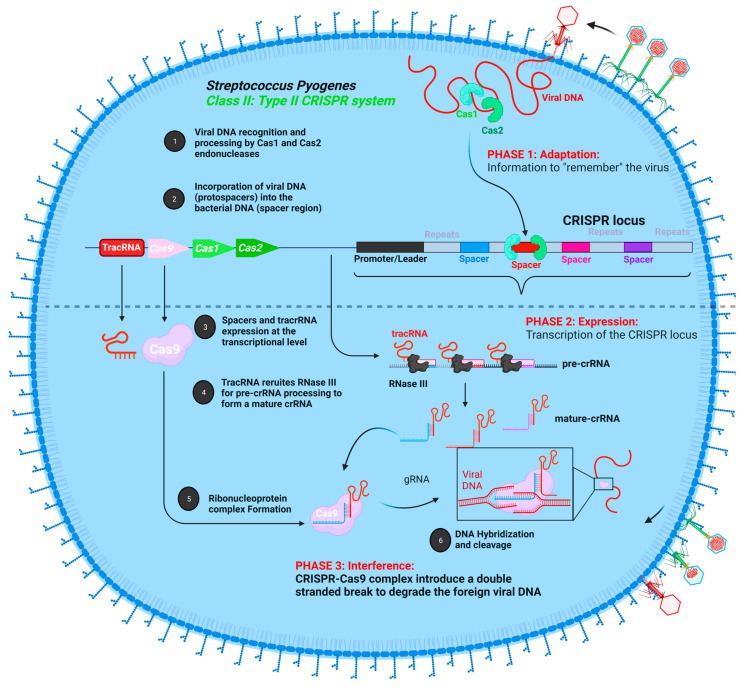
**Key features of CRISPR-Cas9-based acquired immune response in *Streptococcus pyogenes*.** Adaptation. The first step involves the recognition and cleavage of the foreign viral DNA by specific Cas proteins (Cas2 and Cas3) into smaller fragments and their incorporation into the spacer region of the bacterial genome. Expression. The second step involves the expression of the CRISPR locus to generate pre-cRNA, which is afterwards processed by specific RNases (RNase III) to generate mature crRNA. Interference. The third step involves the assembly of the mature crRNA with the tracRNA, which recruits the Cas9 endonuclease to form the CRISPR-Cas9 complex system. Taking the advantage of crRNA’s homology with the viral DNA sequence, the target viral DNA is chopped by Cas9 endonuclease to inactive the virus.

**Figure 4 cells-11-03615-f004:**
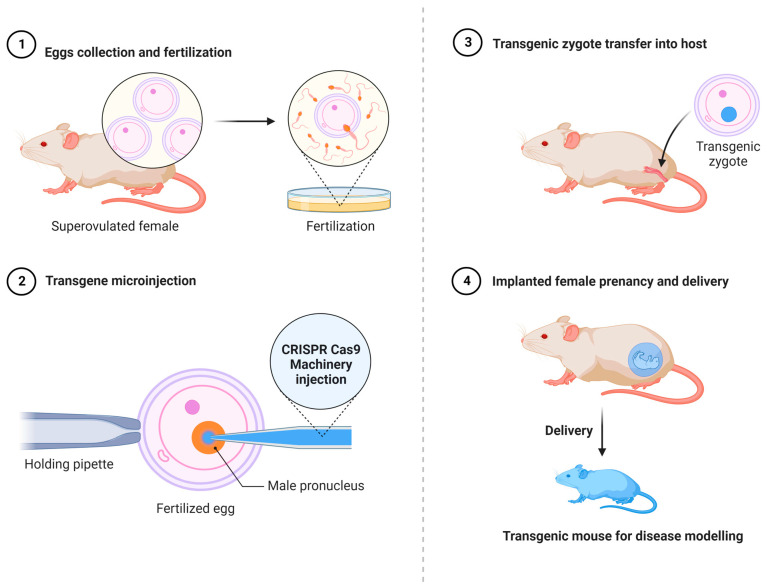
**Direct injection of Cas9 mRNA and sgRNAs into single-cell mouse embryos for genome editing.** The generation of transgenic mouse models can be achieved by collecting eggs from a super-ovulated female mouse followed by in vitro fertilization by the sperms of a male mouse. The fertilized egg then undergoes a direct injection of the CRISPR-Cas9 machinery. After several days, the developed transgenic zygote is reintroduced into a host female mouse to become pregnant and produce a transgenic mouse utilized for modeling the disease of interest.

**Figure 5 cells-11-03615-f005:**
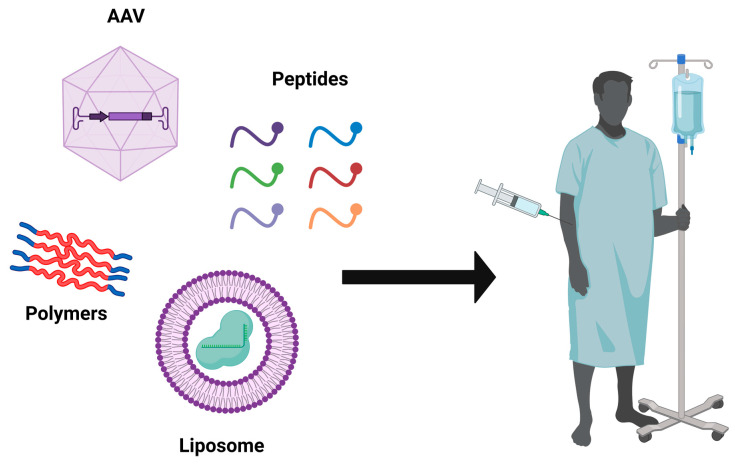
**Viral and non-viral in vivo genome editing strategies for therapeutic applications based on CRISPR-Cas9 machinery.** Viral strategies such as adeno-associated virus and non-viral strategies such as liposomes, polymers, and peptides can deliver CRISPR-Cas9 machinery for in vivo genome editing for therapeutic applications.

**Figure 6 cells-11-03615-f006:**
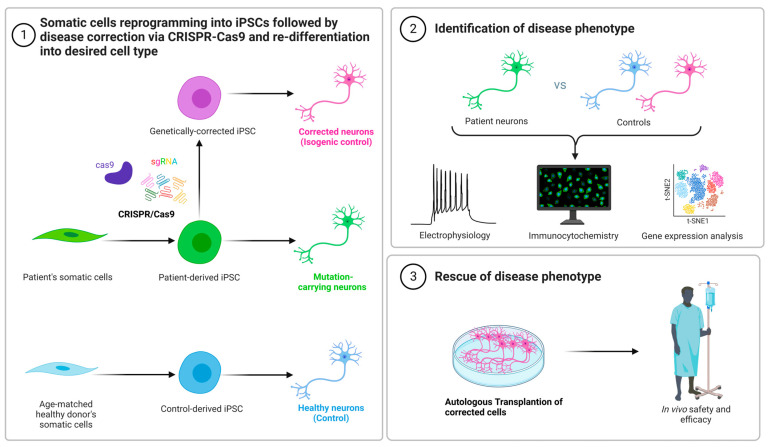
Use of patient-derived induced pluripotent stem cells (iPSCs) with CRISPR-Cas9 for pediatric neurological disorder modeling and for treatment. Patient’s somatic cells can be reprogrammed to generate iPSCs where they can be genetically modified and corrected by CRISPR-Cas9 followed by their re-differentiation into corrected neurons. Characterization of the disease phenotype can be achieved by various techniques including immunocytochemistry, electrophysiology, and gene expression analysis. These corrected neural cells can rescue the disease phenotype of the patient by undergoing autologous transplantation of corrected cells.

**Figure 7 cells-11-03615-f007:**
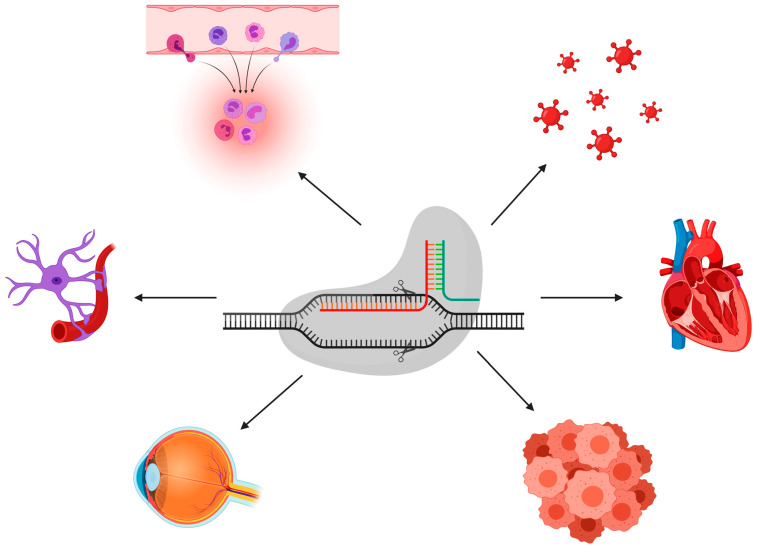
**Various disease modeling applications achieved by CRISPR-Cas9 technology.** CRISPR-Cas9 can be utilized to model various diseases including neurological, immunological, infectious, cardiac, ocular, and cancerous diseases.

**Figure 8 cells-11-03615-f008:**
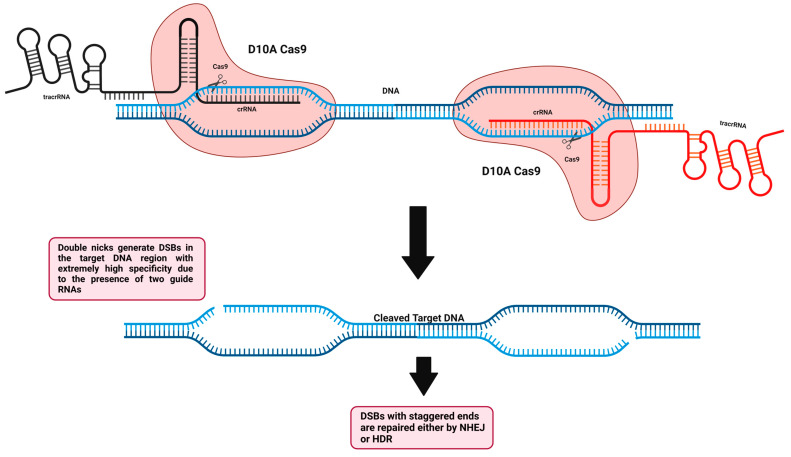
**Cas9 nickase is a variant of the wild-type CRISPR-Cas9 system-based genome editing.** The Cas9 nickase (Cas9n) harboring a mutation in either HNH (H840A) or RuvC (D10A) domains can generate a single nick instead of a DSB at the target site of interest. Dimeric Cas9n with two different guide RNAs can be fused to generate double nicks, one at each part of the target DNA, thus yielding DSBs with staggered ends implementing enhanced specificity with extremely reduced off-target effects.

**Figure 9 cells-11-03615-f009:**
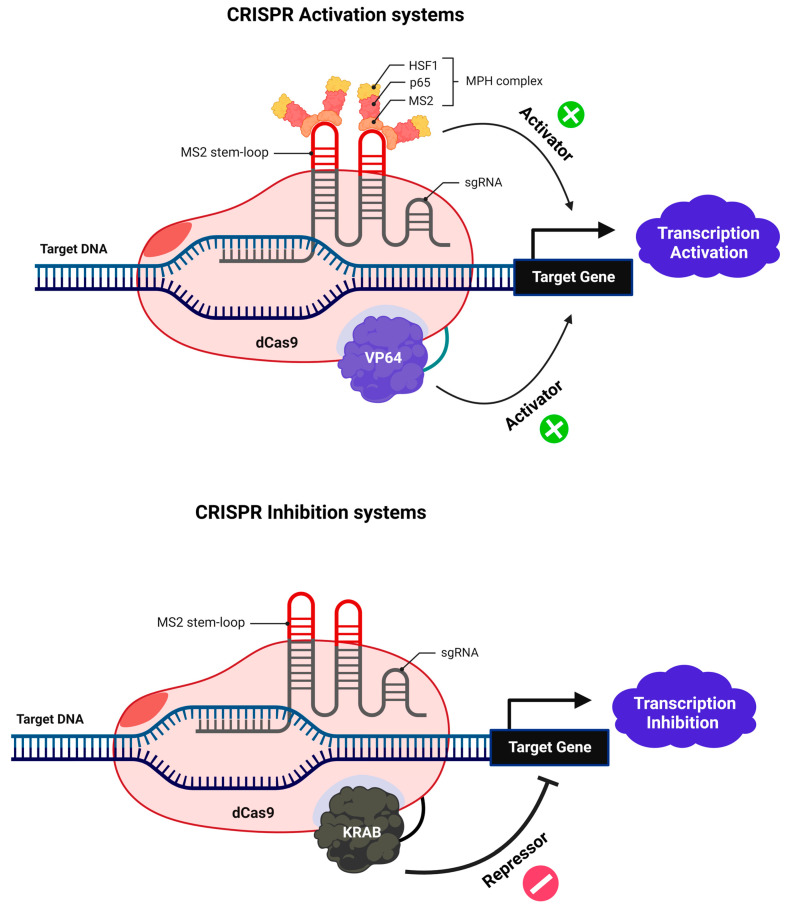
dCas9 is a variant of the wild-type CRISPR-Cas9 system acting as a key modular system for the recruitment of transcriptional regulators. dCas9, which is catalytically inactive, can be utilized for transcriptional regulation processes. dCas9 can recruit various transcriptional regulators acting either positively (activators) or negatively (repressors) at the transcriptional level. For instance, VP64 fused to dCas9 can activate transcription, whilst for KRAB, it can inhibit transcription.

**Figure 10 cells-11-03615-f010:**
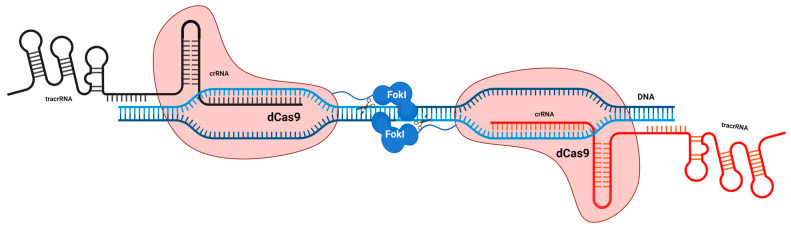
**Dimeric RNA-guided *Fok*I nucleases (RFNs) is a variant of the wild-type CRISPR-Cas9 system-based genome editing.** To ensure highly efficient specificity with reduced off-target events, RFNs were constructed by merging dCas9 coupled to *Fok*I with a guide RNA, which is catalytically inactive and can be utilized for transcriptional regulation processes. dCas9 can recruit various transcriptional regulators acting either positively (activators) or negatively (repressors) at the transcriptional level. For instance, VP64 fused to dCas9 can activate transcription, whilst for KRAB, it can inhibit transcription.

## Data Availability

Not applicable.

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
