# Peer review of "CRISPR-Cas9 Technology for the Creation of Biological Avatars Capable of Modeling and Treating Pathologies: From Discovery to the Latest Improvements"

_cells, 2022, doi:10.3390/cells11223615_

Round 1

Reviewer 1 Report

This paper described three mainly genome-editing tools’ mechanism and application, especially in gene therapy. It also discussed the challenges of CRISPR-Cas9 technology. The organization and writing of this paper are well done, and I think it can be published in Cells with a minor revision.

1.  I think if the authors could add some advancements of CRISPR-Cas9 technology, such as base editors and prime editing, the manuscript will be better. 

2.  At the 6.4 section “CRISPR-Cas9 PAM dependency”, the authors only mentioned Cas9 orthologs to relax the PAM dependent. Some Cas9 variants could also relax the PAM limitation, such as xCas9, Cas9-NG, SpgCas9 and SpRY. I recommend the authors could add this part.

Author Response

Dear Reviewer,

First of all, I would like to thank you very much for your valuable comments, which were really quite beneficial to add more value to the review article. I kindly attach to you a word document with figures with high quality (ameliorated), and also the responses marked in red in the sections of Cas9 nickase and PAM dependency (already added directly) with references highlighted in red in the references section. Benefiting from your expertise, I kindly attach to you the directly the response here also

  1. I think if the authors could add some advancements of CRISPR-Cas9 technology, such as base editors and prime editing, the manuscript will be better. 

Response: 

Nickases can be useful tools for prime and base editors [181], which have the potential to edit the DNA sequence in a precise fashion without the need to introduce double stranded breaks or donor DNA templates, thus no reliance on HDR pathway. Nowadays, base editors comprise a CRISPR-Cas9 nickase (incapable of forming DSBs due to a catalytic domain impairment) linked/fused to either proteins capable of triggering DNA repair or single-stranded DNA deaminase enzyme. Base editors can be classified currently into two major classes: adenine base editors (ABEs), which have the capacity to convert A•T base pairs-to-G•C base pairs, and cytosine base editors (CBEs), which have the capacity to convert C•G base pairs to T•A base pairs [199,200]. Nearly 30% of human disease variants can be due the four transition mutations (G→A, A→G, T→C, C→T) [201]. Both base editors (CBEs and ABEs) can revert or install transition mutations and it was feasible in various cellular models, animal harboring human genetic diseases [202-204]. For an increased precision, prime editing was introduced recently as an effective genome editing technology that is capable of performing the whole 12 possible types of point mutations (insertions, deletions, and substitutions), as well as small regional insertions and deletions in a targeted, and precise manner in the DNA of interest. Prime editors are composed of a CRISPR-Cas9 nickase domain linked/fused to a reverse transcriptase functional domain. This system also requires a uniquely engineered prime editing guide RNA (pegRNA), which has a spacer sequence that can guide the primer editor to the genomic DNA to be precisely edited. Furthermore, it has the designed edit of interest to be added in an extension region at the 3′ end. Following the nicking (single stranded break) of the target DNA by cas9 nickase. The prime editor acquiring the reverse transcriptase functional domain utilizes the newly released 3′ end of the target DNA region to prime/initiate reverse transcription by using the extension region, which is present in the pegRNA as a template. Following reverse transcription, the newly edited DNA target site will exist as a 3’ DNA flap, which is redundant with a 5’ DNA flap having the unedited DNA sequence. Under normal conditions, DNA repair systems tend to remove the 5’ DNA flap, enhancing the incorporation of the edited 3’ DNA flap into the DNA target site, thus generate a heteroduplex DNA where one strand is edited and the other strand remains unedited. To fully install the edit permanently, with the help of a simple sgRNA to further direct the primer editor to induce another nick to the non-edited strand, thus allowing DNA repair systems to faithfully use the edited strand as a template to re-synthesize the non-edited strand, thus generating a permanent edited DNA duplex [188]. Several human cell lines [205], human induced pluripotent stem cells [206], mouse cortical neurons [206], and mouse embryos [207] were utilized for prime editing.

  1. At the 6.4 section “CRISPR-Cas9 PAM dependency”, the authors only mentioned Cas9 orthologs to relax the PAM dependent. Some Cas9 variants could also relax the PAM limitation, such as xCas9, Cas9-NG, SpgCas9 and SpRY. I recommend the authors could add this part.

Response:

It is worth mentioning that the PAM dependency can be positively enhanced with the generation of Cas9 variants like xCas9, which can recognize many PAM sites, including GAA, GAT, and NG in mammalian cells [221], SpCas9-NG is designed to recognize NG sites [222], SpG variant can also easily target a wide set of NGN PAMs and can offer a versatile flexibility in genome editing, SpG variant was further modified and optimized to generate a near-PAMless-SpCas9 variant termed SpRY which can nearly target almost all PAMs including NRN and to a certain limit NYN PAMs [223,224]. Hopefully, with the new cas9 variants, robust activities can be achieved at a higher resolution level.

Wishing you all the best,

Reviewer 2 Report

In this review article W. Rachidi and colleagues present a thorough overview of current genome and epigenome editing tools and their critical contribution to the development of disease models and to advancing therapeutic approaches. The review is extensive and comprehensive, of interest for the scientific community and the amount of graphical description by the authors is highly commendable. 

Minor comments:

·       In the introduction there is a logical discontinuity in line 41 where the authors go from describing the importance and implementation of genome editing for functional genomics and regenerative medicine to the DNA damage response mechanisms. I believe the review could benefit from a more extensive expansion of the first rather than the latter since this is the main subject of the article.

·       In the section starting from line 308 it would be beneficial to go over not only the correction of patient derived iPSCs but also the generation of disease related mutations to develop such systems as an introduction to the following paragraphs.

·       In Figure 5, in vivo delivery of genome editing tools through adenoviral vectors should be presented and discussed in the text.

·       In the “CRISPR-Cas9 and gene therapy section”, the authors should reference ongoing and completed clinical trials.

·       In the “Challenges of CRISPR-Cas9 and enhanced specificity” the authors should comment on the possibility and occurrence of immune reaction to Cas9 in the event of in vivo genome editing.

Author Response

Dear Reviewer,

First of all, I would like to thank you very much for your valuable comments, which were really quite beneficial to add more value to the review article. I kindly attach to you a word document with figures of high quality (ameliorated), and also the responses marked in green in the sections of introduction, Disease modeling and gene therapy using CRISPR-Cas9 (I added a paragraph about the importance of AAVs, and recombinant AAVs and also I ameliorated the section of iPSCs by taking your valuable advice), and in the section of CRISPR-Cas9 and gene therapy, I added some references of clinical trials published, and lastly added a paragraph about the possible host's immune response following CRISPR-Cas9 in vivo delivery 

 Benefiting from your expertise, I kindly attach to you directly the response here also

  1.  In the introduction there is a logical discontinuity in line 41 where the authors go from describing the importance and implementation of genome editing for functional genomics and regenerative medicine to the DNA damage response mechanisms. I believe the review could benefit from a more extensive expansion of the first rather than the latter since this is the main subject of the article.

Response: 

I tried to establish a smooth link to pave the way for the DNA damage concept discussion.

Following extensive attempts, researchers finally relied on a clue to manipulate the genome which is inducing ‘‘DNA damage’’

2. In the section starting from line 308 it would be beneficial to go over not only the correction of patient derived iPSCs but also the generation of disease related mutations to develop such systems as an introduction to the following paragraphs.

Response: 

(iii) Using gene editing in conjunction with human induced pluripotent stem cells (iPSCs) to create models of genetically complicated diseases, given their potential of differentiation to any cell type of interest, if the appropriate cocktail mix is available and for future treatment strategies by correcting these disease models and reintroducing them to the patient. It is feasible to examine human genome alterations in a variety of genetic backgrounds using this method. In vitro disease pathogenesis modeling can be mimicked using iPSCs from patients that have been differentiated in culture to identify disease-affected cells. In iPSCs isolated from diseased people, CRISPR can be utilized to reverse specific mutations, explaining the consequences of such mutations and giving proof of principle for gene therapy (Figure 6). It seems inevitable that CRISPR-Cas9 technology will find its way into therapeutic applications, given the intense interest. 

3. In Figure 5, in vivo delivery of genome editing tools through adenoviral vectors should be presented and discussed in the text

Response: 

(ii) In vivo gene editing involves the delivery of the CRISPR-Cas9 machinery system into desired cells of interest in their natural tissue niches (Figure 5). CRISPR-Cas9 components can be delivered as either polymers, liposomes, peptides, or packed in viruses. The introduction of viral vectors, primarily adeno-associated viruses (AAVs) which are non-enveloped, single stranded DNA viruses, taking advantage of their small size, and non-pathogenic behaviour has enabled in vivo edition [107,108]. These viruses nowadays are becoming more popular to be used as vectors for therapies [109]. Nearly 80% of the human population are seropositive for AAVs, and to date, they are not linked to any type of human disease. Because of their low cytotoxicity, immunogenicity, and the very low chance of integration into the human genome, it is currently on the top list leading in vivo delivery systems for the CRISPR components compared to other viral vectors [110,111]. Another key advantage is the presence of several well-known AAV serotypes for a specific and different tissue tropism, such as for heart cells, epithelial lung cells, neurons, and skeletal muscle cells, making them very efficient for an effective directed tissue-specific delivery for in vivo genome editing [112]. As mentioned previously, AAVs have a low risk of integrating into the human’s genome, which can be to a certain extent a barrier to novel therapeutic applications. Researchers ameliorated those vectors to generate a new variant termed recombinant AAV (rAAV) vector, which can solve the problem of unintended genome integration since they are designed without the implication of Rep genes which are involved in the modulation of viral gene expression, viral DNA replication, and the integration mechanism compared to AAV which has these genes [113,114]. Currently, in preclinical trials, the delivery of CRISPR-Cas9 machinery via the recombinant AAV vector has effectively shown an alleviation in atherosclerosis disease in mutant mice for low density lipoprotein receptor (LDLR) [115]. Furthermore, the Food and Drug Administration (FDA) has approved the utilization of AAV vectors in hundreds of clinical trials [116]. AAV delivery system has shown its potential to treat various in vivo disease models including muscular dystrophy [117,118], neurodegenerative diseases [119], sickle cell anemia [120], and metabolic liver diseases [121]. 

  4. In the “CRISPR-Cas9 and gene therapy section”, the authors should reference ongoing and completed clinical trials.

Manipulation of DNA or RNA for the treatment or prevention of human diseases is known as gene therapy. By disrupting endogenous disease-causing genes, correcting disease-causing mutations, or introducing new protective genes, precise genome editing holds the potential to permanently cure a wide variety of life-threatening diseases. Various clinical trials were launched to treat blood disorders [177], cancers [178-180], and currently, other ongoing trials to treat eye diseases, chronic infections, and rare protein folding diseases. For gene therapy applications, CRISPR-Cas9 provides a unique, highly effective genome editing technique.

5. In the “Challenges of CRISPR-Cas9 and enhanced specificity” the authors should comment on the possibility and occurrence of immune reaction to Cas9 in the event of in vivo genome editing.

5.6. Immune response against Cas9 endonuclease

The two major sources of Cas9 endonuclease are Streptococcus pyogenes and Staphylococcus aureus, which are well known for causing human diseases including strep throat and MRSA. Nearly 80% of the healthy population individuals have shown to develop a cellular, T-cell mediated and a humoral, antibody mediated immunity against both bacteria. The human’s immune system acts as a protective shield against exogenous surface and secreted proteins. So, the question will highlight the fact that Cas9 is classified as an intracellular protein, what will be the future for in vivo Cas9-mediated gene editing for novel human disease therapies?[226]

I already added the references in the word document in their appropriate positions (highlighted in green)

I would like to thank you very much for all your efforts and advices!

Wishing you all the best,
